# Plasma Functionalization of Multi-Walled Carbon Nanotubes for Ammonia Gas Sensors

**DOI:** 10.3390/ma15207262

**Published:** 2022-10-18

**Authors:** Alexander G. Bannov, Anton M. Manakhov, Dmitry V. Shtansky

**Affiliations:** 1Department of Chemistry and Chemical Engineering, Novosibirsk State Technical University, 20 K. Marx, 630073 Novosibirsk, Russia; 2National University of Science and Technology MISIS, Leninsky Prospekt 4, 119049 Moscow, Russia; 3Research Institute of Clinical and Experimental Lymphology—Branch of the Institute of Cytology and Genetics, Siberian Branch of Russian Academy of Sciences, 2 Timakova st., 630060 Novosibirsk, Russia

**Keywords:** carbon nanotubes, functionalization, plasma, gas sensor, ammonia, response, plasma treatment, functional groups

## Abstract

The role of plasma functionalization of multi-walled carbon nanotubes (MWCNTs) for room-temperature ammonia gas sensors was investigated. Plasma functionalization of MWCNTs with maleic anhydride was carried out at various durations. The active material of the gas sensor was investigated by scanning electron microscopy, transmission electron microscopy, Raman spectroscopy, and X-ray photoelectron spectroscopy. It was shown that the formation of functional groups on the surface of carbon nanotubes led to an increase in the ammonia sensor response by two to four times. The increase in functionalization duration induced the rise of O/C from 0.28 to 0.335, an increase in sensor resistance, and the distortion of the shape of the I-V curves.

## 1. Introduction

Carbon nanotubes (CNTs) have attracted interest within the last few decades and research into their synthesis, modification, and applications is still being carried out throughout the world. Multi-walled carbon nanotubes (MWCNTs) are a promising material for various applications due to their lower cost and simplicity of synthesis and scale-up compared to single-walled carbon nanotubes. MWCNTs can be used in polymer composites [1,2], hydrogen evolution reactions [3,4], electronics [5], biosensors [6], etc.

In addition, CNTs can be used as active materials for gas sensors [7,8,9]. Recently, research into room-temperature gas sensors based on CNTs for ammonia detection has been of particular interest [10,11,12]. However, the response of gas sensors based on untreated CNTs is relatively low. Thus, there are several approaches to enhance the sensing properties, such as the use of conducting polymers [13], plasma treatment [14,15], and the deposition of metallic [16] and oxide [10,17] nanoparticles. Plasma functionalization significantly improves the capture of ammonia molecules on the surfaces of carbon nanotubes. In [18], oxygen plasma was used for the modification of multi-walled carbon nanotubes, and the sensitivity of the modified gas sensors towards ammonia was twice as high as that of the untreated sensing material. The combination of oxygen plasma treatment of MWCNTs and deposition of maleic anhydride (MA) and acetylene plasma co-polymers [19] led to the formation of core-shell structures with a relatively high response (22.5% towards 100 ppm NH_3_). Kim et al. [15] created an ammonia gas sensor based on an O_2_-functionalized MWCNT/PANI sensor, reaching a response of 3.34%/(ppm NH_3_) within a range of 10–100 ppm NH_3_.

In this paper, the effect of the duration of plasma functionalization of MWCNTs using co-polymerization of maleic anhydride and acetylene for an ammonia gas sensor was investigated. This approach led to enhancement of the MWCNTs sensor response towards NH_3_ thanks to the increased concentration of surface functional groups through changing the duration of the plasma treatment.

## 2. Materials and Methods

### 2.1. Synthesis of MWCNTs

MWCNTs were deposited onto Si/SiO_2_ substrate by plasma-enhanced chemical vapor deposition (PECVD) (Figure 1a). The growth of CNTs was carried out on the iron catalytic nanoparticles formed as a result of the decomposition of Fe(CO)_5_. The iron oxide nanoparticles were deposited in a microwave plasma torch with a dual-flow nozzle electrode according to the technique described in [20,21]. Si/SiO_2_ substrate was placed in a holder for 4 samples, with a 4 × 4 mm deposition area for each sample. Argon was used as a carrier gas. The argon flow rates through the central and outer channels were 700 and 28 sccm, respectively. The outer channel was used to feed the Fe(CO)_5_ vapors. The deposition of nanoparticles was carried out during 15 s at a plasma power of 210 W.

The Si/SiO_2_ substrates with iron nanoparticles were placed into a tubular furnace, where the decomposition of acetylene was performed at 600 °C. Firstly, the iron oxide catalytic nanoparticles on the Si/SiO_2_ substrates were reduced in an Ar (1400 sccm)/H_2_ (500 sccm) mixture for 10 min; then, the flow was switched to Ar (1400 sccm)/C_2_H_2_ (25 sccm) and the deposition of MWCNTs was carried out for 10 min. In the first stage, hydrogen reduces the iron oxide nanoparticles to iron ones, and in the second stage the decomposition of acetylene led to the formation of carbon on them. The carbon deposits in the form of carbon nanotubes and carbon nanofibers (formed predominantly on the large-sized catalytic nanoparticles). Finally, the carbon coating on the Si/SiO_2_ substrate formed. Then, to achieve a good electrical contact between parts of carbon active material, the gold contacts were deposited using the low-pressure PVD method. The active layer of sensors was covered by gold contacts (325 nm layer, 6.65 × 2.33 mm) with a 25 nm-thick NiCr interlayer. The latter was used for better adhesion between the gold layer and the substrate.

### 2.2. Plasma Treatment

MWCNT-based sensors were plasma-treated in MA-C_2_H_2_ plasma using the equipment described in [22]. Briefly, the sensor with sealed Au contacts was placed into a special holder (Figure 1b). The deposition of the plasma polymer was carried out by dielectric barrier discharge (DBD) plasma co-polymerization of MA and C_2_H_2_. The discharge was ignited between planar metallic electrodes covered by Al_2_O_3_ ceramics, 1 mm-thick, on the bottom of which the sensor was placed. The gases were supplied by a 1 mm slit between the high-voltage electrodes. The vapors of MA were fed into the discharge by using Ar flow (Messer, Bad Soden, Germany, 99.998%) purging through the MA pellets (Sigma-Aldrich, Burlington, MA, USA, 99%). The flow rate of the second monomer, C_2_H_2_ (Messer, Bad Soden, Germany, 99.6%), was set to 2 sccm. The Ar flow rate through the bubbler was 50 sccm. The total Ar flow rate was kept at the level of 150 sccm. The deposition was carried out in the discharge ignited by 5.4 kHz sinusoidal high voltage. The power supplied by the generator was 20 W. Three samples of gas sensors with various durations of deposition (2, 5, and 7 min) were obtained.

### 2.3. Testing of Ammonia Gas Sensors

The chemiresistive gas sensor was characterized via measuring the resistance of the sensor under ammonia exposure. The measurements were performed using a custom-made setup with two gas channels: synthetic air (Linde, Dublin, Ireland) and the analyte (NH_3_ diluted in the air). The scheme of the setup is described in [19] in detail (also shown in Figure 1c). The sensor was placed into the measuring chamber on the heater, operated by an Agilent U3606A DC power supply (Agilent, Santa Clara, CA, USA). The resistance of the active layer of the sensor was measured by the two-point technique between the gold electrodes using the Keithley 2410 Source Meter (Keithley, Solon, OH, USA). A fixed-bias voltage of 1 V was applied. The sensor response was calculated as:
(1)$$\frac{\Delta R}{R_{0}}=\frac{R-R_{0}}{R_{0}}$$
where *R* and *R*_0_ are the resistances of sensors in a mixture of synthetic air/NH_3_ and in pure synthetic air, respectively. The concentration of ammonia was varied from 100 to 500 ppm. All samples were tested at the same conditions: room temperature (25 ± 2 °C) and relative humidity 2–2.5%.

### 2.4. Characterization of Samples

Current–voltage (I–V) characteristics of the sensors were measured by the 4200-SCS Semiconductor characterization system (Keithley, Solon, Ohio) within a range from −5 to +5 V at room temperature (25 °C). The morphology of the CNT-based sensing material was determined by scanning electron microscopy (SEM) using Tescan MIRA (TESCAN, Brno, Czech Republic) equipped with an electron-dispersive X-ray spectroscopy add-on. The disorder degree of CNTs was determined by Raman spectroscopy using a Renishaw spectrometer (Renishaw, Wotton-under-Edge, UK) in a range of 1000–1800 cm^−1^ (λ = 514 nm).

X-ray photoelectron spectroscopy (XPS) was used to analyze the chemical composition of surface layers using an EA125 spectrometer fitted on a custom-built UHV system. The measurements were used at a pass energy of 25 eV and power of 270 W. To excite XPS spectra, Al Kα radiation (1486.6 eV) was used.

## 3. Results and Discussion

SEM images of the MWCNTs (after functionalization) used as an active material for ammonia gas sensors are depicted in Figure 2a–c. TEM images of the initial non-treated MWCNT sample are shown in Figure 2d–f. As can be seen, the morphology of carbon nanotubes remains almost unchanged after plasma functionalization with maleic anhydride because of the low duration of treatment and relatively mild plasma conditions. The samples of MWCNTs deposited directly on the Si/SiO_2_ substrate were represented by strongly entangled carbon nanotubes. The sample also contained a small fraction of cup-stacked carbon nanofibers and chain-like carbon nanofibers (Figure 2d–f) [23], formed as a result of growth of carbon on the large-size (above 20–30 nm in diameter) catalytic iron nanoparticles.

Raman spectra of plasma-treated MWCNTs are shown in Figure 3a. The spectra of plasma-functionalized MWCNTs consist of two bands typical for carbon nanomaterials, D mode (1353 cm^−1^), which can be attributed to the disordered structure, and G mode (1589–1592 cm^−1^), relating to the graphitic structure of the material [24,25]. The ratio of the intensity of these two modes indicates the disorder degree of the material. Plasma treatment leads to an increase in I(D)/I(G) (Table 1), indicating a decrease in the contribution of C-C bonds in the form of sp^2^ hybridization. It can be noted that any treatment of MWCNTs can produce many defects on its surface, but the results on I(D)/I(G) confirm the relatively mild conditions of plasma functionalization compared to other types of treatment, such as chemical functionalization, as reported in [26]. Plasma treatment can be considered as an alternative way compared to liquid-phase chemical functionalization, which is accompanied by considerable losses of material during oxidation [27] since the yield of CNT material is almost 100%. The attachment of functional groups formed on the surface of carbon nanotubes passes only on the surface, with no etching of the surface layer, and therefore, no release of CO_x_ formed as a result of the oxidation of carbon.

I–V curves of the active layer of ammonia gas sensors are presented in Figure 3b. The current–voltage (I–V) characteristic of the sensor treated for 2 min was almost linear, which corresponds to the conductive material. Increasing the treatment time led to the appearance of non-linearity of the I-V curve due to the change in the conduction mechanism to semiconducting because of the functional coating formed on the carbon surface. Sensor responses are shown in Table 2.

The resistance of sensors is rising with the increasing duration of treatment due to the formation of a plasma coating on the surfaces of MWCNTs. This complements the data on Raman spectroscopy and XPS. The resistance of the initial, non-treated MWCNTs was only 18 kΩ, and the plasma treatment led to an increase in resistance, to 52.7–110 kΩ. This was caused by the formation of a relatively thin film of plasma polymer on the surfaces of MWCNTs. This film possesses a higher resistance compared to carbon nanotubes and acts as an additional contact resistance between nanotubes, changing the paths for the transport of charge carriers.

Typical curves of the sensor response are shown in Figure 3c. Generally, the resistance of the sensor grows under contact with ammonia. This is a typical mechanism of ammonia’s interaction with carbon nanomaterials. NH_3_ is an electron-donor compound, which donates electrons during adsorption on the surface of carbons. These electrons decrease the concentration of charge carriers in MWCNTs and the resistance of the latter increases [28,29]. This shows that the charge carriers in MWCNTs studied in this paper are holes. It is worth noting that the plasma functionalization did not change the mechanism of response of gas sensors, just the ΔR/R_0_ value changes.

The response of gas sensors was within the range of 0.3–0.6% for 100–500 ppm NH_3_. The plasma treatment led to the growth of the sensor response to 1.7–2.9% (2 min). The maximum response was observed at 5 min of treatment, and further growth of the duration of treatment led to a decrease in response, to 0.7–1.3% (100–500 ppm). This is in agreement with the O/C determined by XPS, which was 0.28, 0.35, and 0.335 for 2, 5, and 7 min, respectively. XPS shows no nickel on the spectra, confirming the preservation of carbon shells on the iron nanoparticles, which is a considerable difference compared to liquid-phase chemical treatment [30]. For further surface characterization, the XPS C1s curve fitting was performed. The XPS C1s signals for all samples were fitted using the same model, composed of hydrocarbons C-C/C-H (BE = 285 eV, used for calibration), carbon singly bonded to oxygen C-O (BE = 286.5 ± 0.1 eV), carbon doubly bonded to oxygen C=O/O-C-O (BE = 287.9 ± 0.1 eV), and carboxylic acid or ester group C(O)O (BE = 289.2 ± 0.15 eV).

All XPS spectra of the treated samples are presented in Figure 4. The data show that the increase in the duration of treatment above some level (5 min) led to the formation of a lower concentration of surface oxygen-containing functional groups, especially C(O)O. The increase of the concentration of carboxylic groups from 2 to 5 min induced the growth of the NH_3_ sensor response. Further growth of the duration of treatment led to a decrease in the concentration of carboxyls and an increase of phenolic groups (carbon single-bonded to oxygen). This fact indicates the dominant role of carboxylic groups in the formation of the response of CNTs to NH_3_.

Some increase in the response time may be attributed to the creation of an additional functional coating on the surface of carbon nanotubes, preventing the transportation of charge carriers from the surface of materials under contact with NH_3_. The dynamics of the formation of this coating can be estimated by the values of resistance shown in Table 2.

Considering the response time of sensors, it can be noted that the response time (i.e., the time which is necessary to reach 90% of the final response value) slightly changes after functionalization of MWCNTs (Appendix A). For example, the response time for the non-treated sensor was 390, 407, and 349 s at 100, 250, and 500 ppm of ammonia, respectively. The treatment led to a small change of this value, and according to the error of its determination, it can be considered as insignificant (399, 409, and 358 s). The values of response times were comparable with data reported for plasma-treated core-shell nanocarbon structures [21]. Usually, the increase of the resistance of sensors is higher [31,32] compared to the data presented in this paper, but this depends on the type of sensor, the thickness of the film, and the design of the measurement cell (dimensions, volume), and therefore it is difficult to compare the data.

## 4. Conclusions

The presented results show that plasma functionalization of MWCNTs can be effective in the enhancement of ammonia detection using room-temperature gas sensors. Plasma functionalization made it possible to increase the sensor response towards ammonia by 3–4 times compared to pristine MWCNTs. The increase in the duration of plasma functionalization induced the increase of O/C from 0.28 to 0.335, an increase in sensor resistance, and distortion of the shape of the I-V curves. Plasma treatment of MWCNTs can be considered as among the most intensive types of treatment (zero-loss treatment), which maintains the material and makes it possible to achieve almost 100% yield of the material treated.

## Figures and Tables

**Figure 1 materials-15-07262-f001:**
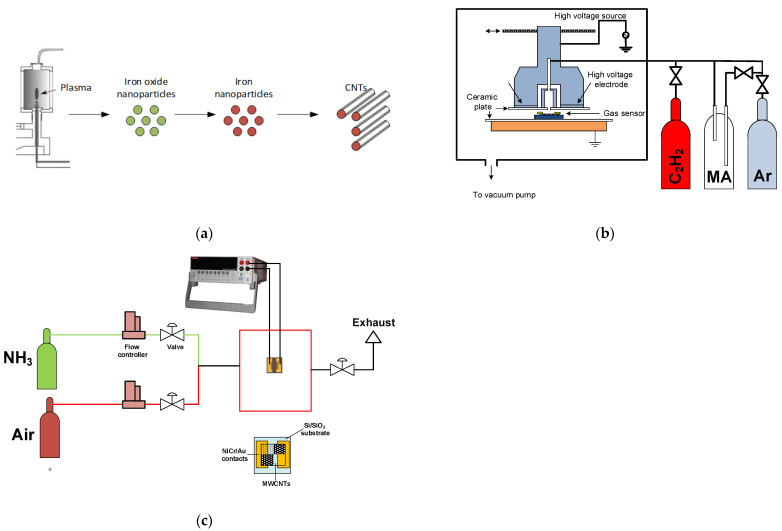
Scheme indicating all stages of synthesis of MWCNTs (**a**). Scheme of experimental setup for plasma treatment of MWCNTs on Si/SiO_2_ substrates (**b**). Scheme of equipment used for testing of gas sensors (**c**).

**Figure 2 materials-15-07262-f002:**
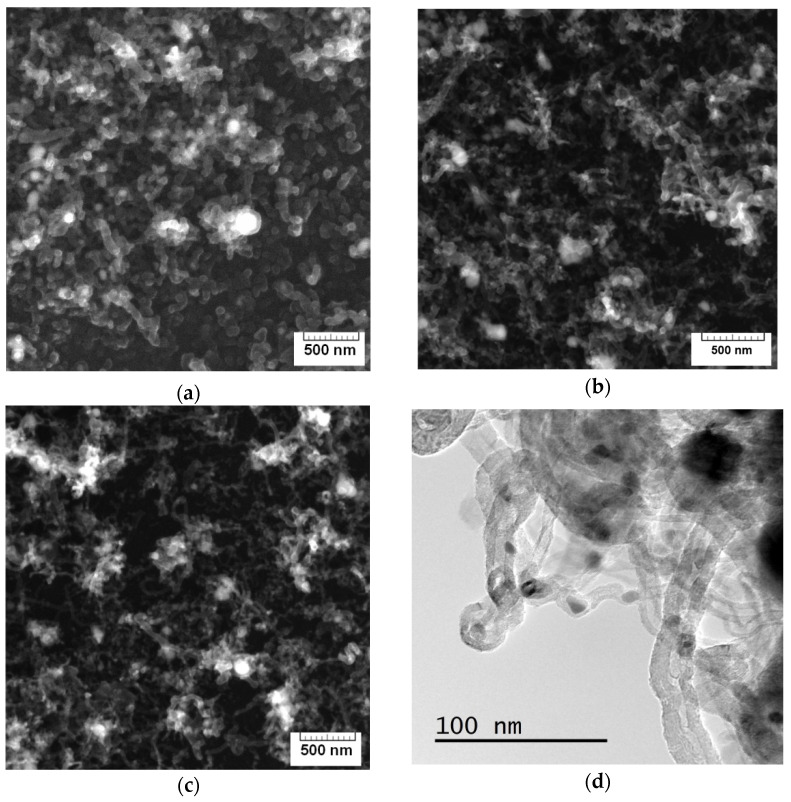
SEM images of MWCNTs after functionalization at various durations: (**a**) 2 min, (**b**) 5 min, and (**c**) 7 min. (**d**–**f**) TEM images of MWCNTs synthesized (non-treated sample).

**Figure 3 materials-15-07262-f003:**
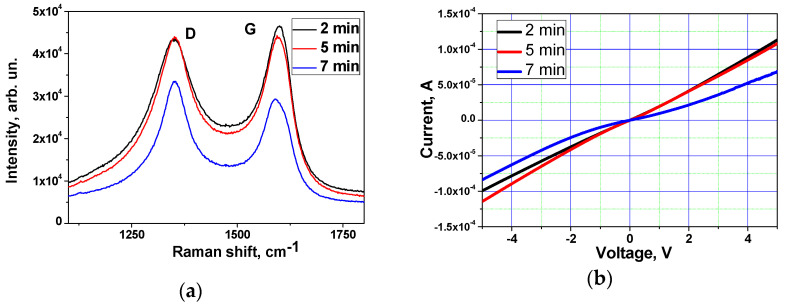
(**a**) Raman spectra of samples obtained using various durations of plasma treatment. (**b**) I-V curves of the active layer of the gas sensor. (**c**) Typical response curve of pristine MWCNTs and MWCNTs plasma-treated for 2 min.

**Figure 4 materials-15-07262-f004:**
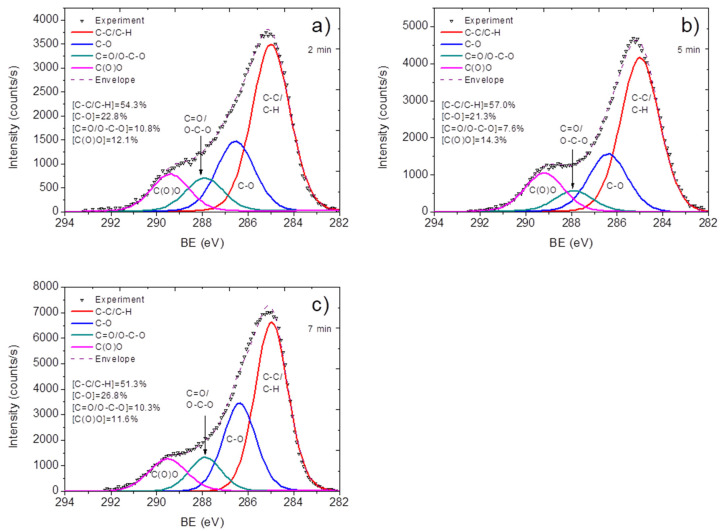
XPS spectra of MWCNT samples after being plasma-treated for 2 min (**a**), 5 min (**b**), and 7 min (**c**).

**Table 1 materials-15-07262-t001:** Parameters of Raman spectra of plasma-treated MWCNTs.

Sample	D Peak Position, cm^−1^	D Peak FWHM, cm^−1^	G Peak Position, cm^−1^	G Peak FWHM, cm^−1^	I(D)/I(G)
Non-treated	1348	131	1590	69	0.86
2 min	1353	130	1591	65	0.87
5 min	1353	119	1592	70	0.95
7 min	1353	90	1589	59	1.08

**Table 2 materials-15-07262-t002:** Response of gas sensors before and after functionalization of the MWCNT active layer.

Duration of Treatment	Resistance of Sensor, kΩ	Sensor Response before Functionalization, %			Sensor Response after Functionalization, %		
		100 ppm	250 ppm	500 ppm	100 ppm	250 ppm	500 ppm
2 min	52.7				1.7	2.3	2.9
5 min	58.6	0.4	0.6	0.7	1.8	2.6	3.6
7 min	110				0.7	1.0	1.3

## Data Availability

Data are available from the corresponding author upon reasonable request.

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
