# Peer review of "Plasma Functionalization of Multi-Walled Carbon Nanotubes for Ammonia Gas Sensors"

_materials, 2022, doi:10.3390/ma15207262_

Round 1

Reviewer 1 Report

Authors presented the multi-walled carbon nanotubes (MWCNTs) treated by plasma for different duration from 2 to 7 min. The results showed that the plasma functionalization process could increase the responsivity for ammonia sensors. The manuscript has well described the data and procedures for these sensors. The reasons have also been given in this article. Some presentations could be improved    before publication by this journal, which are listed below.

1.      Authors could add a schematic figure to explain their equipment for measuring the responsivity of ammonia gas sensors in section 3.3. testing of ammonia gas sensors.

2.      Figure 2 showed the responsivity in dialt(R)/R (%), the time dependent curves indicate an up going trend. I suggested authors could add more discussions about this.

3.      The response times (rising and fall times) should be estimated and compared with other published articles.

Author Response

First of all, the authors are very grateful to the reviewer for his/her valuable comments and essential suggestions. We have made significant changes in our manuscript while keeping all modifications tracked. Please find below our answers in the point/by-point style.

"Authors presented the multi-walled carbon nanotubes (MWCNTs) treated by plasma for different duration from 2 to 7 min. The results showed that the plasma functionalization process could increase the responsivity for ammonia sensors. The manuscript has well described the data and procedures for these sensors. The reasons have also been given in this article. Some presentations could be improved    before publication by this journal, which are listed below."

  1. Authors could add a schematic figure to explain their equipment for measuring the responsivity of ammonia gas sensors in section 3.3. testing of ammonia gas sensors.

Answer:

Figure 1 has been added in order to show the equipment for testing of gas sensors.

  1. Figure 2 showed the responsivity in dialt(R)/R (%), the time dependent curves indicate an up going trend. I suggested authors could add more discussions about this.

Answer:

The short description of mechanism of ammonia detection was added.

“Generally, the resistance of the sensor grows under contact with ammonia. This is a typical mechanism of ammonia interaction with carbon nanomaterials. NH3 is an electron-donor compound, which donating electrons during adsorption on the surface of carbons. These electrons decreases the concentration of charge carriers in MWCNTs and the resistance of latter increases [28,29]. This shows that the charge carriers in MWCNTs studied in this paper are holes. It is worth noting that the plasma functionalization did not change the mechanism of response of gas sensors, just the ΔR/R0 value changes.”

  1. The response times (rising and fall times) should be estimated and compared with other published articles.

Answer:

The response time strongly depends on the design (size) of the cell for sensor testing and on the value of the total flow rate in a cell. Therefore this value must be varied from one paper to another, and it is not correct to compare them. Anyway, the table with the data on response time is presented in the Supplementary materials. The description of the change in response time of sensors is also given in the paper revised.

“One consider the response time of sensors, it can be noted that the response time (i.e. time which is necessary to reach the 90% of final response value) slightly changes after functionalization of MWCNTs (Table 1S in Supplementary materials). For example, the response time for non-treated sensor was 390 s, 407 s, and 349 s at 100 ppm, 250 ppm, and 500 ppm of ammonia, respectively. The treatment led to small change of this value and according to error of its determination can be considered as insignificant (399 s, 409 s, and 358 s). The values of response time was comparable with data reported for plasma treated core-shell nanocarbon structures [21].

Some increase in response time may be attributed to the creation of additional functional coating on the surface of carbon nanotubes, preventing the transportation of charge carriers from the surface of materials under contact with NH3. The formation dynamics of this coating can be estimated by the value of resistance shown in Table 2.”

Since the recovery of sensors is incomplete it is not correct to analyze these values and compare the data reported with the sensors with better recovery, especially for CVD graphene, thin films of MXenes, and others, which possess a better response since the thickness of their active layer is low enough.  

Reviewer 2 Report

The SEM images are of poor quality, improved images should be provided. In the text it is mentioned that Figure 1d, which is indicated as displaying the TEM image of plasma synthesized CNTs, shows no evidence of nanofibers. The authors should clarify this or provide another image showing this clearly. 

Author Response

First of all, the authors are very grateful to the reviewer for his/her valuable comments and essential suggestions. We have made significant changes in our manuscript while keeping all modifications tracked. Please find below our answers in the point/by-point style.

The SEM images are of poor quality, improved images should be provided. In the text it is mentioned that Figure 1d, which is indicated as displaying the TEM image of plasma synthesized CNTs, shows no evidence of nanofibers. The authors should clarify this or provide another image showing this clearly. 

Authors Response:

  1. The SEM images with better quality were added to the paper.
  2. The additional TEM images were added to Figure 2. They show that the sample consists not only of nanotubes, some of the large nanoparticles of catalysts that take part in the growth of carbon nanofibers. Their diameter of them is higher than 30-40 nm and their channel is very narrow (compared to MWCNTs) some of them can be considered chain-like structure carbon nanofibers.

Reviewer 3 Report

The paper is a good next step in the development of  ammonia sensors based on MWCNTs.

Author Response

The paper is a good next step in the development of  ammonia sensors based on MWCNTs.

Response:

The authors thank reviewer for the valuable comments that we implemented in the revised version. All changes are highlighted.

Reviewer 4 Report

In this investigation, MWCNTs were subjected to plasma functionalization for different durations, and the effect of duration on the magnitude of ammonia sensor response was studied. I notice some sections are written and discussed satisfactorily, but some sections are very confusing. I provided my suggestions and comments below. With these changes, the manuscript is acceptable for publication.

1.     Page 2, line 53

Section 3.1 (wrong subsection number, should be 2.1

2.     Authors wrote

“The nanoparticles were deposited in a microwave plasma torch with dual….”

Nanoparticles of what? Please explain.

3.     Synthesis of MWCNTs section needs more clarity. This section is very confusing.

4.     On Page 3, lines 113, authors wrote

The samples of MWCNTs deposited directly on the Si/SiO2 substrate were represented by strongly entangled carbon nanotubes with a large number of walls.

I do not see large number of wall !! This is confusing…

5.     Authors wrote:

SEM images of the MWCNTs (before and after functionalization) used as an active material for ammonia gas sensors are depicted in Figure 1a-c.

Which one is before and which one is after functionalization? Same issue in the figure caption of figure 1.

Figure 1. SEM images of MWCNTs before and after functionalization at various durations: (a) 2 117 min, (b) 5 min, (c) 7 min; (d) TEM images of MWCNTs synthesized subjected to plasma treatment.

6.     Fix spacing related issue in

(BE=286.5±0.1 eV)

(BE=287.9±0.1 eV)

Fix such errors throughout the article

7.     I suggest to depict images for section   “ Synthesis of MWCNTs” and “Plasma treatment”

Author Response

First of all, the authors are very grateful to the reviewer for his/her valuable comments and essential suggestions. We have made significant changes in our manuscript while keeping all modifications tracked. Please find below our answers in the point/by-point style.

In this investigation, MWCNTs were subjected to plasma functionalization for different durations, and the effect of duration on the magnitude of ammonia sensor response was studied. I notice some sections are written and discussed satisfactorily, but some sections are very confusing. I provided my suggestions and comments below. With these changes, the manuscript is acceptable for publication.

  1. Page 2, line 53

Section 3.1 (wrong subsection number, should be 2.1

 ANSWER:

Thank you for your remark. The corrections were made.

  1. Authors wrote

“The nanoparticles were deposited in a microwave plasma torch with dual….”

Nanoparticles of what? Please explain.

ANSWER:

An explanation of iron oxide nanoparticles is given.

  1. The synthesis of MWCNTs section needs more clarity. This section is very confusing.

 ANSWER:

The technique of synthesis of MWCNTs was extended.

  1. On Page 3, lines 113, authors wrote

The samples of MWCNTs deposited directly on the Si/SiO2 substrate were represented by strongly entangled carbon nanotubes with a large number of walls.

I do not see large number of wall !! This is confusing…

 ANSWER:

The part of the sentence concerning the walls was deleted.

  1. Authors wrote:

SEM images of the MWCNTs (before and after functionalization) used as an active material for ammonia gas sensors are depicted in Figure 1a-c.

Which one is before and which one is after functionalization? Same issue in the figure caption of figure 1.

Figure 1. SEM images of MWCNTs before and after functionalization at various durations: (a) 2 117 min, (b) 5 min, (c) 7 min; (d) TEM images of MWCNTs synthesized subjected to plasma treatment.

ANSWERs:

Thank you for your remarks. The correct Figure and Figure captions are presented.

Figure 2. SEM images of MWCNTs after functionalization at various durations: (a) 2 min, (b) 5 min, (c) 7 min; (d-f) TEM images of MWCNTs synthesized (non-treated sample).”

The SEM images were presented for treated samples since TEM will destroy the active layer of material. TEM images were shown for the sample before functionalization because there are more informative. Plasma treatment did not change the diameter and length of nanotubes, the SEM images of the functionalized samples are the same, and looks the same as non-treated initial sample.

  1. Fix spacing related issue in

(BE=286.5±0.1 eV)

(BE=287.9±0.1 eV)

Fix such errors throughout the article

 ANSWERs:

This has been fixed.

  1. I suggest to depict images for section “Synthesis of MWCNTs” and “Plasma treatment”

 ANSWERs:

The Figures (Figures 1a-c) were added to clarify the equipment for synthesis, plasma treatment of MWCNTs as well as testing of sensors.

Round 2

Reviewer 4 Report

The changes are very satisfactory, and therefore, I recommend publishing the work.